Using natural landscape and instream habitat to identify stream reference groups for bioassessment

Dyer Joseph J. 1 joseph.dyer@conservation.ok.gov
Dvorett Daniel 1
Flotemersch Joseph 2
1 Water Quality, Oklahoma Conservation Commission , Oklahoma City, OK , United States
2 Independent Researcher , Fort Thomas, Kentucky , United States
Steinke Dirk
Electronic publication date: 2025 Oct 21
Publication date: 2025
Volume: 13
Electronic Location ID: e20234
Received 2025 Jul 11; Accepted 2025 Sep 23
Copyright: © 2025 Dyer et al.
Copyright year: 2025
Copyright holder: Dyer et al.
License: This is an open access article distributed under the terms of the Creative Commons Attribution License, which permits unrestricted use, distribution, reproduction and adaptation in any medium and for any purpose provided that it is properly attributed. For attribution, the original author(s), title, publication source (PeerJ) and either DOI or URL of the article must be cited.
License URL: https://creativecommons.org/licenses/by/4.0/

Keywords: Ecological archetypes, Bioassessment, Multimetric indices, Reference groups

Funding: The authors received no funding for this work.

==============================
Background

Grouping streams into reference groups based on their similarities is a critical component of developing multi-metric indices to evaluate biotic integrity. The use of level III ecoregions is a common approach that has been successful for many geopolitical regions. However, the diversity in ecoregions across the state of Oklahoma results in excessive complexity in the application of reference groups. In this study, we sought to simplify Oklahoma’s reference groups by considering the natural drivers of species distribution in wadeable streams.

Methods

We used two K-means clustering algorithms to create hierarchical stream groups. In the first K-means analysis, we grouped wadeable streams using several climatic and geologic features expected to influence fish distributions across the state (Tier I). Next, we subdivided the stream groups identified in the Tier I analysis based on water chemistry and instream habitat (Tier II). We used classification trees and between class multivariate analyses to validate and understand the resulting groups and define them as ‘stream archetypes’ based on distinguishing habitat characteristics.

Results

The two K-means analyses resulted in six groups that represented both regional differences and reach-scale geomorphological differences. We determined that there were two regions, West and East that were primarily distinguished by mean annual precipitation. Tier II clustering identified Plains and Valley stream groups in both the East and West regions. These groups represented shallow, run-dominated, sand bed streams, and relatively deep, pool dominated, silty streams, respectively. The East region possessed two additional stream groups (i.e., Hills and Highlands groups), each with rocky substrate and riffle channel units that broke up the runs and pools in the stream. Of these, the Hills group was more prone to having pools than runs and had low nutrient concentrations compared to the Highlands group. We found that the six groups explained variability in habitat and water chemistry as well as the previous 13 ecoregion-based reference groups. Further, we confirmed that the groups were not the result of common anthropogenic influences on stream ecosystems. The six groups were subsequently organized into three stream archetypes, Plains (East and West), Valley (East and West) and Rocky (Hills and Highlands). We expect these six hierarchical stream groups and three archetypes to be useful in evaluating biotic communities in Oklahoma moving forward.

Introduction

The index of biotic integrity has served as an important method of bioassessment for nearly four decades and has become a primary tool for assessing an ecosystem’s ability to sustain ecological function in the context of local and landscape degradation. Karr et al. (1986), proposed the Index of Biotic Integrity (IBI), a type of bioassessment, which evaluated the composition of taxa, trophic guilds, and the abundance and condition of individuals in the context of region-specific expectations. While Karr et al. (1986) first promoted the use of IBIs in lotic ecosystems for fish assemblages, IBI development has since expanded into a variety of aquatic habitats (i.e., rivers, lakes and wetlands) and biological communities; macroinvertebrates (e.g., Klemm et al., 2003), macrophytes (e.g., Beck et al., 2010), and algae (e.g., Wang & Stevenson, 2005; Wu, Schmalz & Fohrer, 2012; Zalack, Smucker & Vis, 2009). Additionally, IBI methodology has been implemented around the globe including in South America (Bozzetti & Schulz, 2004), Europe (Angermeier & Davideanu, 2004), Africa (Kamdem-Toham & Teugels, 1999), Asia (Jia, Sui & Chen, 2013), and Australia (Harris & Silveira, 1999). Over the last 40 years of IBI development, biological communities have been continuously found to respond in predictable ways to complex patterns of chemical pollution and habitat degradation. As a result, bioassessment has become a reliable and cost-effective method to identify the most prominent threats to ecosystem condition and address problems promptly before more extensive degradation occurs (Karr & Chu, 1999).

Since biological communities vary with biogeography, IBI development and application typically occurs within regions of relatively homogenous physical environment, so variation within metrics can be attributed to anthropogenic disturbance (Barbour et al., 1996). Around the world it is common to develop IBIs that are specific to discrete river basins (e.g., Aparicio et al., 2011; Harris & Silveira, 1999; Qadir & Malik, 2009 and Zhu et al., 2023). Among streams in the United States, a combination of Omernik (1987) level III ecoregions and measures of stream size or water temperature appear to be common grouping mechanisms (e.g., Lyons, Wang & Simonson, 1996; McCormick et al., 2001). Ecoregions have served as a logical and convenient mechanism to group streams for IBIs, because ecoregions are defined by the climate, geology, flora, and fauna that exist within them. However, overlaying ecoregional boundaries with political jurisdictions such as states, where uniform bioassessment methods are needed to support water quality standards, can be problematic. In fact, 31 states contain at least six ecoregions, necessitating the development of multiple reference datasets. Further, approximately 20% of state/ecoregion combinations occupy less than 2,500 km2, often making identification of appropriate regional reference conditions infeasible. Ecoregion based reference conditions have been particularly complicated in Oklahoma, where there are 12 level III ecoregions and two disjunct geologic formations (Arbuckle Uplift and Wichita Mountains, level IV ecoregions) despite the relatively small spatial extent of the state (Omernik, 1987).

A common approach to reduce spatial complexity is to aggregate ecoregions based on similarities in fish and macroinvertebrate assemblages. For example, the >30 ecoregions across 10 western states were condensed to three major regions with 10 sub-ecoregions based on best professional judgement of stream types (Whittier et al., 2007). For the United States Wadeable Streams Assessment (WSA), Herlihy et al. (2008) used 2-dimensional nonmetric multidimensional scaling ordination to aggregate 84 national ecoregions into nine large ecoregions based on relative abundances of macroinvertebrates at reference sites. The choice of ecoregion size was constrained to ensure that enough reference sites existed within each region, but at the potential trade-off of greater within-region variability. The trade-off between IBI regionalization scale and reference site variance should be evaluated by considering application goals. Where reference site variability may lead to greater error bars on population-level condition class categorization, it may lead to incorrect water-quality standards-support decisions for site-specific assessment of waterbodies. In 2020, a collaborative attempt by TetraTech and multiple Oklahoma agencies to use aggregated ecoregions to update fish IBIs for Oklahoma for water quality standard support, was unable to both adequately reduce natural variability and retain sufficient reference sites within groups. Streams in centrally located ecoregions were transitional in their fish community and ecoregions that had more homogeneous fish communities were smaller and did not have sufficient sites in disturbance categories to complete model validation (Jessup, 2020).

For site-specific bioassessment of streams, alternative categorization approaches beyond ecoregions may help reduce the natural variability in biological communities. While communities are affected by climatologic and geologic variables that determine ecoregion boundaries, more local variables remain unaccounted for with ecoregions alone. Stevenson (1997) described ultimate, intermediate, and proximate drivers of species distribution where ultimate drivers include climate and geology, which in turn drives intermediate features (e.g., flow regime and stream morphology), resulting in the observed habitat, flora, and fauna composition within a channel unit (proximate features). A common approach is incorporating measures of stream size in the creation of IBI metrics because of the numerous natural gradients (e.g., slope, width, organic inputs, temperature, etc.,) that exist along the stream continuum (Vannote et al., 1980; Fausch, Karr & Yant, 1984; Rahel & Hubert, 1991). However, the existence of these gradients within ecoregions further complicates the comparison of streams (Hughes & Gammon, 1987; McCormick et al., 2001; Klemm et al., 2003). Small ecoregions may already consist of a small sample population, and when the complexity of the stream continuum is considered, it can result in an inadequate number of reference sites that represent the natural variability among high-quality streams.

The goal of grouping streams for the development of an IBI is to account for expected differences in the taxa composition among various regions in the monitoring jurisdiction. Therefore, stream groups for reference criteria should consider the most salient natural components of the physical environment and habitat that predict species distribution (Karr & Chu, 1999). Predictions of species distribution benefit from hierarchy theory in which broad-scale (e.g., landscape) features restrict the range of potential fine-scale (reach or channel unit) features (Frissell et al., 1986; Allan, 2004; Allen & Hoekstra, 2015). Therefore, as a study group is established, the set of relevant environmental variables that impact biological integrity is ‘cut’ at a particular hierarchy. Bedoya, Manolakos & Novotny (2011) posit that stressors just below the ‘cut line’ will be the most impactful drivers of biological community response, while those above the ‘cut line’ are relatively homogenous within the region. By using a hierarchical approach to group streams based on physical attributes, we attempt to a priori control for natural variation among landscape scale variables such as climate and geology, and intermediate features such as stream morphology. As a result, variability within a group should primarily result from stressors that affect the proximate or instream conditions at the scale of our data collection. By controlling for both naturally occurring ultimate and intermediate features, we hypothesize a stronger link between observed impairments and anthropogenic activities that occur heterogeneously within the landscape.

Although stream classification based on the drivers of species occurrence is an established concept, potential pitfalls with the approach exist. First, Karr & Chu (1999) warned against the over classification of habitats where the number of habitat types was greater than the number of possible fish assemblages, and it is instead better to choose the fewest number of classes necessary. Second is the potential pitfall of inadvertent grouping of sites based on a habitat feature that is the result of poor land use management. For instance, if conventional row crop management results in streams with homogenous streambeds of fine sediment and increased nutrient loads (Cooper, 1993; Lewandowski & Cates, 2023), creating a group using these streams would result in a distinct, more lenient, set of standards for streams with poor management practices. Moreover, if land managers in the watershed began to implement practices like no-till or cover crops (Sharpley & Smith, 1991), or riparian improvement (Pusey & Arthington, 2003), we would not be able to detect any improvement in conditions, because the stream was already meeting the inappropriately defined standards prior to conservation efforts.

The goal of this article is to simplify the current ecoregion groupings in Oklahoma by accounting for multiscale, natural drivers of species distribution in support of identifying and setting reference criteria useful in the assessment of condition and management of Oklahoma’s wadeable streams. The three objectives of this study were: 1. determine the fewest number of stream-type groups based on watershed- and reach-scale physicochemical features, 2. verify that the new stream classification scheme explains natural variability in physicochemical features as well as or better than current ecoregion groups, and 3. confirm that none of our groups are a result of anthropogenic stressors on the landscape. We hypothesize that these three objectives will collectively create more useful groups for reference conditions while avoiding the pitfalls discussed above.

Materials and Methods

Study area

The Oklahoma state boundaries served as the extent of the study area. Oklahoma is in the central United States and is a transitional area between the humid Gulf Coastal Plains, and the semi-arid Central Great Plains regions. There is a strong precipitation gradient from the western portion of the state that commonly receives <500 mm of rainfall annually and the eastern portion that may receive >1,300 mm of rainfall in the same year (Oklahoma Climatological Survey, 2021). Additionally, from the southeast corner of the state to the northwest corner, the landscape gains >1,400 m of elevation (Mesonet, 2025). As would be expected from the elevation gradient, rivers in Oklahoma generally flow from the northwest to the southeast; however, the varied topography of the Eastern third of the state (i.e., Ozark Highlands, Boston Mountains, and Ouachita Mountains) provides some exceptions to the general direction of flow. The physical-chemical composition of the landscape mirrors the precipitation gradient. For example, conductivity and alkalinity measurements are orders of magnitude higher in the western extremes compared to the eastern extremes. As previously mentioned, there are 12 level III ecoregions in the state and 2 level IV ecoregions (Fig. 1) that have been determined ecologically meaningful (OCC Oklahoma Conservation Commission, 2022). The High Plains level III ecoregion does not receive enough water consistently enough to support monitoring sites, so the Oklahoma standards are based on 13 modified ecoregions (OCC Oklahoma Conservation Commission, 2005).

Figure 1 Map of the state with ecoregions.

The state of Oklahoma with the ecoregions that are currently used to group wadable streams for Index of Biotic Integrity assessments. The ecoregions data are available at https://catalog.data.gov/dataset/level-iv-ecoregions-of-oklahoma12 (accessed: 9/12/2025).

Data

We included data collected between 2001 and 2019 by the Oklahoma Conservation Commission’s (OCC) Rotating Basin Monitoring and Blue Thumb programs (RBMP and BT, respectively) along with publicly available data in this study. The RBMP began in 2001 and is responsible for monitoring nonpoint source pollution in accordance with section 319 of the Clean Water Act (United States Code, 2018). With the RBMP, OCC collects habitat and water chemistry in wadeable streams (i.e., streams that may be surveyed without a boat at base flow) across the state of Oklahoma. The state is divided into five major planning basins that are each monitored for two consecutive years. The RBMP starts and ends the monitoring of basins in a staggered fashion so that two basins are monitored in any given year, and the entire state is covered once every 5 years. Within a 5-year monitoring cycle, physical and chemical water properties were recorded approximately once every 5 weeks for 2 years and habitat was surveyed once along a 400-m reach. The BT program organizes volunteers around the state and assists them with monitoring their local streams. Although habitat was surveyed in the same manner as in the RBMP, the water chemistry data at BT sites were not. Because BT data were more representative of urban streams than RBMP data and we intend to assess BT fish data with the same methods as RBMP data, we included BT data in the broad-scale analysis where chemistry was not required. The collected data are assessed by OCC and reported to the United States Environmental Protection Agency (USEPA). Publicly available data were obtained from StreamCat (USEPA, 2025). StreamCat is a product of the USEPA National Aquatic Resources Survey and contains 600 GIS—derived metrics at the catchment and watershed scales.

Ideally, the environmental features included in this study would be temporally stable and unaffected by human activities but given the 18-year time frame that was unlikely (Ward, 1989). As such, landscape features included in the study with greater temporal variability were included as long-term averages where data availability was sufficient (e.g., annual rainfall). Reach-scale habitat features were expected to vary among the 5-year monitoring cycles, particularly due to the multi-year droughts that occurred during the study period (Oklahoma Climatological Survey, 2025). Because extreme low flows reduce habitat availability, habitat surveys conducted during a drought year could misrepresent the typical stream conditions. Therefore, we did not aggregate habitat surveys among multiple monitoring cycles and instead treated them separately so that anomalous surveys could be detected. Finally, conductivity and alkalinity were influenced by underlying geology across the state but could potentially be acutely affected by anthropogenic activities. Therefore, we used the median of the measurements collected within a monitoring cycle to mitigate potential extreme values.

K-means cluster analysis

We used unsupervised K-means clustering to create K-means groups of physically similar wadeable streams in Oklahoma. All statistical analyses were conducted with R Statistical software (vers. 4.3.1, R Core Team, 2023). The K-means cluster method is like multivariate ordination methods in that all features provided to the algorithm and their correlations are considered equally (Kumar et al., 2011; Krishna, 2015). One potential limitation of the K-means clustering method is the assumption of spherical equal-variance clusters which could result in the misclassification of sites that better fit an elongated distribution (Jain, 2010). Any classification scheme of continuous data requires arbitrary delineations to be made (Cullum et al., 2017) and because we could test the assumption with several post hoc tests we were comfortable with this assumption. Other limitations to the K-means clustering method are the effects of choosing the correct number of clusters and the random selection of cluster centroids (Krishna & Murty, 1999). We used the ‘NbClust’ (vers. 3.0.1, Charrad et al., 2014) package to iterate between two to 15 groups and calculate 30 separate indices on each grouping (Charrad et al., 2014). When an index was optimized, the corresponding number of groups received a “vote” in favor of that index. A “vote” is a positive expression in favor of a particular number of groups being optimal. We considered the grouping number with the most “votes” from all indices to be optimal, visualized the resulting clusters in multivariate space via the ‘factoextra’ package (Kassambara & Mundt, 2020), and assigned a K-means group to sample sites. The spatial distributions of K-means groups were viewed in QGIS (vers. 3.32.0, QGIS Development Team, 2023).

Another potential limitation to the K-means cluster method as it pertains to ecology is the inability to account for hierarchical relationships. To incorporate hierarchy theory into our clustering, we first created clusters with only climate and geology features assessed at the watershed and catchment scales (Tier I clusters). Climate and geology features included in the first iteration of clusters were chosen based on their known relationships to species distributions (e.g., average annual rainfall, elevation) or the correlation to water quality parameters that are likely linked to fish distributions (e.g., Sulphur was correlated with conductivity and Calcium Oxide was correlated with alkalinity). Water table depth and soil permeability were included at this scale because they vary regionally and potentially influence the ground water availability during dry-weather periods. For the second iteration (Tier II clusters) we separated the K-means groups (regions) defined by the first cluster iteration. At the reach scale, we included watershed area and sinuosity features as these features are related to flow regime (Dodds, 2002). Instream habitat features from OCC’s habitat surveys were selected to represent streambed morphology, which is a product of long-term flow regimes (Dodds, 2002), and channel unit-level habitat that influence species composition (Table 1). We then performed the same analysis as before on each subset of streams with the fine-scale features.

Table 1 Table of environmental features and the sources used in this study.

Analysis	Feature	Description	Source	
T1 CLUSTER	Elevation	Mean elevation of catchment	StreamCat (NHDPlus)	
T1 CLUSTER	Slope	Mean slope of catchment	StreamCat (NHDPlus)	
T1 CLUSTER	PermCat	Mean permeability (cm/hour) of soils (STATSGO) within catchment	StreamCat (STATSGO_Set2)	
T1 CLUSTER	RockDep	Mean depth (cm) to bedrock of soils (STATSGO) within catchment	StreamCat (STATSGO_Set2)	
T1 CLUSTER	WTDep	Mean seasonal water table depth (cm) of soils (STATSGO) within catchment	StreamCat (STATSGO_Set2)	
T1 CLUSTER	AnRainC	PRISM climate data-30-year normal mean precipitation (mm): Annual period: 2008 and 2009 within catchment	StreamCat (PRISM_0809)	
T1 CLUSTER	CaOWs	Mean % of lithological calcium oxide (CaO) content in surface or near surface geology within watershed	StreamCat (GeoChemPhys1)	
T1 CLUSTER	SWs	Mean % of lithological sulfur (S) content in surface or near surface geology within watershed	StreamCat (GeoChemPhys1)	
T2 CLUSTER/BCA	Sin	Sinuosity of the sample reach; straight-line distance between start and end points divided by 400 m sample reach	OCC Habitat Surveys	
T2 CLUSTER/BCA	WS_Area	Area (km2) of watershed upstream of sample location	OCC Habitat Surveys	
T2 CLUSTER/BCA	Rocky.Cover	Percent of habitat attributed to bedrock, boulder, cobble, and gravel	OCC Habitat Surveys	
T2 CLUSTER/BCA	Woody.Cover	Percent of habitat attributed to submerged woody debris	OCC Habitat Surveys	
T2 CLUSTER/BCA	Veg.Cover	Percent of habitat attributed to submergent, emergent, and submerged terrestrial vegetation	OCC Habitat Surveys	
T2 CLUSTER/BCA	Silt	Percent of substrate at 20-m transects comprising silt or unconsolidated clay substrate (diameter < 0.2 mm)	OCC Habitat Surveys	
T2 CLUSTER/BCA	Sand	Percent of substrate at 20-m transects comprising sand substrate (diameter 0.2–2 mm)	OCC Habitat Surveys	
T2 CLUSTER/BCA	Gravel	Percent of substrate at 20-m transects comprising gravel substrate (diameter 2–64 mm)	OCC Habitat Surveys	
T2 CLUSTER/BCA	Cobble	Percent of substrate at 20-m transects comprising cobble substrate (diameter 64–250 mm)	OCC Habitat Surveys	
T2 CLUSTER/BCA	Boulder	Percent of substrate at 20-m transects comprising boulder substrate (diameter > 250 mm)	OCC Habitat Surveys	
T2 CLUSTER/BCA	Bedrock	Percent of substrate at 20-m transects comprising bedrock substrate	OCC Habitat Surveys	
T2 CLUSTER/BCA	W.D	Mean wetted width divided by the mean depth at 20-m transects	OCC Habitat Surveys	
T2 CLUSTER/BCA	Depth.Var	Variance of all depth measurements collected	OCC Habitat Surveys	
T2 CLUSTER/BCA	Max.Pool.Depth	Maximum depth measured in a pool channel unit	OCC Habitat Surveys	
T2 CLUSTER/BCA	Pools	Percent of total channel units categorized as a pool	OCC Habitat Surveys	
T2 CLUSTER/BCA	Runs	Percent of total channel units categorized as a run	OCC Habitat Surveys	
T2 CLUSTER/BCA	Riffles	Percent of total channel units categorized as a riffle	OCC Habitat Surveys	
T2 CLUSTER/BCA	Alkalinity	Median alkalinity measurement	OCC Water Chemistry Surveys	
T2 CLUSTER/BCA	Conductivity	Median conductivity measurement	OCC Water Chemistry Surveys	
BCA	TN	Median total nitrogen measurement	OCC Water Chemistry Surveys	
BCA	Available.N	Median available nitrogen measurement	OCC Water Chemistry surveys	
BCA	TP	Median total phosphorus measurement	OCC Water Chemistry Surveys	
BCA	OP	Median orthophosphate measurement	OCC Water Chemistry Surveys	
BCA	Chloride	Median chloride measurement	OCC Water Chemistry Surveys	
BCA	Sulfate	Median sulfate measurement	OCC Water Chemistry Surveys	
BCA	DO	Median dissolved oxygen (mg/L) measurement	OCC Water Chemistry Surveys	
BCA	DO.perc	Median dissolved oxygen percent saturation measurement	OCC Water Chemistry Surveys	
BCA	pH	Median pH measurement	OCC Water Chemistry Surveys	
BCA	TSS	Median total suspended solids measurement	OCC Water Chemistry Surveys	
BCA	Turbidity	Median turbidity measurement	OCC Water Chemistry Surveys	
BCA	Temperature.Water	Median water temperature (C) measurement	OCC Water Chemistry Surveys	
STRESSOR BCA	OpenUrb	Percent of land cover classified as open development in the catchment	StreamCat (NLCD)	
STRESSOR BCA	LowUrb	Percent of land cover classified as low-intensity development in the catchment	StreamCat (NLCD)	
STRESSOR BCA	MedUrb	Percent of land cover classified as medium-intensity development in the catchment	StreamCat (NLCD)	
STRESSOR BCA	HiUrb	Percent of land cover classified as high-intensity development in the catchment	StreamCat (NLCD)	
STRESSOR BCA	Hay	Percent of land cover classified as hay and pasture in the catchment	StreamCat (NLCD)	
STRESSOR BCA	Crop	Percent of land cover classified as crop cultivation in the catchment	StreamCat (NLCD)	
STRESSOR BCA	CBNF	Mean rate of biological nitrogen fixation from crop cultivation (kg N/ha/yr), within the catchment	StreamCat (AgricultureNitrogen)	
STRESSOR BCA	Fert	Mean rate of synthetic nitrogen fertilizer application (kg N/ha/yr), within the catchment	StreamCat (AgricultureNitrogen)	
STRESSOR BCA	Manure	Mean rate of manure application (kg N/ha/yr), within catchment	StreamCat (AgricultureNitrogen)	
STRESSOR BCA	ICI	Revised Index of catchment integrity (version 2.1)	StreamCat (ICI_IWI_v2.1)	
STRESSOR BCA	CCHEM	Revised index of regulation of water chemistry component score calculated using catchment metrics	StreamCat (ICI_IWI_v2.1)	
STRESSOR BCA	RoadDens	Density of roads (2010 Census Tiger Lines) within catchment (km/square km)	StreamCat (RoadDensity)	
STRESSOR BCA	Road_Xing	Density of roads-stream intersections within catchment (crossings/square km)	StreamCat (RoadStreamCrossing)	
STRESSOR BCA	Houses	Mean housing unit density (housing units/square km) within catchment	StreamCat (USCensus2010)	
STRESSOR BCA	Population	Mean populating density (people/square km) within catchment	StreamCat (USCensus2010)	
Note:

Features included in the analyses performed in this study. StreamCat data are available at https://www.epa.gov/national-aquatic-resource-surveys/streamcat-dataset (accessed 12/13/24) and OCC data are available at https://occwaterquality.shinyapps.io/OCC-app23b/ (accessed 12/13/24).

Comparing stream groups to ecoregion

We labeled and assessed the resulting hierarchical K-means groups (stream groups hereafter). Because instream habitat was assessed at sites on multiple occasions over the past two decades, some sites were assigned a different stream group at different points in time. We addressed this issue by first, assigning the mode of stream-group designation for each site. In some instances, there was no single mode for stream-group designation. To group sites without a single mode, we used a classification tree (created with the rpart package; Therneau & Atkinson, 2022) to designate the appropriate stream group of the non-modal sites. The classification trees provide quantitative descriptor in the form of calculated thresholds in the features that best explain the variance among groups.

Next, we used the classification trees, between class analysis (BCA), and the spatial distribution of the sites in each cluster to understand the differences and label each stream group. The BCA is a special case of principal components analysis (PCA) where a single factor variable is considered as an explanatory variable (Doledec & Chessel, 1989). The BCA provided several post hoc tests on the validity of the resulting groups including Eigen values, cluster visualization, and a permutation test to assess the validity of the clusters (Dray & Dufour, 2007). The permutation test was conducted on the BCA results with 999 replicates and a p-value < 0.05 was considered significant separation. We presented our results to the field staff who each had 15 to 30 years of experience collecting data at the sites we were evaluating. Once we were satisfied that the stream groups had passed the reality test, we moved ahead with further statistical comparison.

We compared the stream groups and ecoregions to understand the differences in instream habitat and water chemistry using a BCA. The stream groups and ecoregions served as the explanatory variables and were compared using BCA provided by the ‘ADE4’ package (Dray & Dufour, 2007). The sites were then plotted in the multivariate space of instream habitat and water-chemistry. To compare habitat among the stream groups, we included the same features that were used in the Tier II cluster analysis. For water chemistry, we used the median value from 20 water chemistry metrics. Median values mitigate acute contamination events, but water chemistry may be chronically influenced by human activity (Allan, 2004). For this reason, water chemistry was not included in the formation of groups, however, viewing water chemistry post hoc allows us to understand how the groups represent the natural gradients that exist across the state (Fukami & Wardle, 2005). We conducted a separate BCA for the stream group and ecoregion variables and reported the percent inertia explained by each analysis as a metric for explanatory power. In the previous BCA assessment, we found that the sole site in the Wichita Mountains ecoregion may not fit with the stream groups in the western half of the state. Therefore, we labeled the Wichita Mountain site as a unique stream group to see where it would plot relative to the other six groups.

Anthropogenic stressors

Finally, to verify that none of our stream groups were the result of systematic anthropogenic stressors, we included several agriculture, development, and catchment integrity features in a BCA. The expectation was that each stream group should span the range of anthropogenic stressors and catchment integrity indices. All sites within a stream group plotting at one end of any feature in multivariate space would indicate that sites within that stream group were the result of a common stressor, and that the stream group was not valid for setting reference standards.

Results

Tier I clusters

The Tier I cluster analysis included only features that described the climate and geology at sample sites. We included 1,876 samples from 1,069 sites in the K-means cluster analysis. The preliminary analysis indicated that either nine groups (seven votes) or two groups (six votes) best explained the data. No other number of cluster groups tested received more than three votes. Because the goal of this project was to simplify our metrics, we determined that the two-group cluster was most appropriate for subsequent analysis. This separated our sites into East and West regions (Fig. 2). The cluster analysis did not provide a specific threshold for any feature that would allow a rule-based distinction between East and West cluster groups. Therefore, we used a classification tree to determine that annual rainfall at the catchment scale was the primary predictor of East and West clusters. We subsequently labeled any sites with <975.5 mm of annual rainfall as West, and sites with ≥975.5 mm of annual rainfall as East (Fig. 3). These two regions served as the basis for our Tier II cluster analysis.

Figure 2 Map of the state with proposed stream groups.

State of Oklahoma with ecoregions and the surveyed sites categorized by the proposed stream archetype classification. The ecoregions data are available at https://catalog.data.gov/dataset/level-iv-ecoregions-of-oklahoma12 (accessed 9/12/2025).

Figure 3 Classifications trees comparing stream groups.

Classification trees used to designate the K-means cluster group of sites that did not have a consistent group designation over time. The East vs. West tree (A) was used for the Tier I designation and annual rainfall at the catchment scale (mm) was the sole discriminator. The East Group tree (B) was used to distinguish the four groups in the East regions where Cobblel was the log-transformed Cobble substrate (%), Runs was the percent of channel units classified as runs, Boulderl was the log-transformed Boulder substrate (%), and W.Dl was the log-transformed width to depth ratio. The West Group tree (C) was used to differentiate between the two Western groups where the percent of channel units classified as Pools was the only discriminator.

Tier II clusters: east

The East region included 1,081 samples at 631 sites from the Tier I analysis. The sample size was reduced to 522 at 258 sites, due to missing water-chemistry data. Forty-five percent of the sites had only been sampled for one monitoring cycle, whereas 24% were sampled in two, 18% sampled in three cycles, and 13% sampled in four cycles. The cluster analysis indicated that the four-group analysis was best with seven votes, and the next closest was two or three groups each with only five votes. Using the modal method of group assignment for each site, 17 (n = 43 samples) sites were still indicated as non-modal.

The stream-group names were based on a combination of classification trees, BCA, and the spatial location of sites within the four proposed stream-groups. First, we used a classification tree to assign 17 non-modal sites to a stream group. The classification tree revealed that large substrate, portion of run channel units, and width-to-depth ratio were the best differentiators of stream groups (Fig. 3). The Highlands group (n = 74 streams) included nearly every site in the Ozark Highlands, Boston Mountains, and Arbuckle Uplift and relatively few in the Flint Hills and Ouachita Mountains. The Highlands were characterized by having primarily gravel substrate and a large portion of run and riffle channel units. Comparatively, most of the streams in the Hills group were in the Ouachita Mountains and the Flint Hills with one site in the Boston Mountains. The Hills group (n = 37) had more boulders and rocky cover with more pools compared to the Highlands. The East Valleys group (n = 112) tended to be nested between the topographical features of the eastern part of the state (e.g., Flint Hills, Ozarks, Boston & Ouachita mountains). The East Valleys were characterized by deep pools with a low width to depth ratio, silty substrate, and ample woody cover. Finally, the East Plains group (n = 35) was common in the transitional Cross Timbers Ecoregion in the central part of the state. The East Plains group was primarily comprised of sandy runs with higher conductivity and alkalinity than other eastern sites giving it some commonality with streams in the West regional group.

Tier II cluster: west

The West regional group consisted of 795 samples from 438 sites. When missing water-chemistry data were excluded, the sample size was reduced to 382 samples across 140 sites. Of those, 24% were sampled in one cycle, 21% in two, 15% in three, and 40% in four monitoring cycles. The analysis for the best number of clusters indicated that two groups were best with six votes. Four or 14 groups received the second most votes at five votes each, whereas all other cluster numbers received two or fewer votes. After applying the modal method of group assignment to our sites we again had 17 non-modal sites (n = 50 samples).

Unlike the East region, there was no apparent spatial distribution to the western sites; however, viewing the classification tree and BCA of the two groups revealed two ecologically different stream types (Figs. 2 and 3). The West Plains group (n = 70) had less pool habitat and more runs, along with more sand substrate, lower depth variability and maximum pool depth, and a greater width to depth ratio. In contrast, the West Valleys group (n = 70) comprised more pool habitat, with silty substrate, and depth metrics were opposite of those in the West Plains. Although the western portion of the state was less topographically varied than the East, the difference in the streams appeared to be driven by elevation, slope, and water velocity. These differences were expected to create different stream ecosystems.

Comparing stream groups to ecoregions

Using the BCA, we confirmed that the six groups had not violated K-means assumptions. Three axes were needed to explain >= 90% of variance (57%, 23%, and 9% for axes 1, 2, & 3, respectively) in conjunction with the plot (Fig. 4) indicated that clusters were not elongated. The permutation test returned a p-value of < 0.01, confirming significant separation among the six groups.

Figure 4 Between class analysis comparing stream groups to physicochemical features.

Between class analysis between stream groups and habitat and water chemistry features. A principal components analysis of habitat and water chemistry features was used to create the multivariate space (A) and the ellipses associated with the six K-means cluster groups and Wichita Mountains (B).

The six stream groups explained the variation in habitat and water chemistry as well as the 13 modified ecoregions and exemplified three stream archetypes (Valley, Plains and Rocky). The inertia score by the six stream groups BCA was 27% compared to 25% inertia score by the 13 modified ecoregions. The upper portion of the multivariate plot (positive loadings on the y-axis) associated with the BCA contained the East Valleys and West Valleys stream groups (Fig. 4). In the Valley archetype, pool-channel units, silty substrate, and woody cover were most common. The West Valleys had saltier and more alkaline water chemistry than the East Valleys. The West Valleys also had more sand substrate, vegetative cover, and were shallower than the East Valleys, giving the West Valleys some commonality with the Plains archetype that plotted in the lower-right quadrant of the multivariate plot. The lower-right quadrant was associated with run-channel units and higher DO values. Unlike the Valleys, the Plains archetype had the least depth variability and the lowest scores for maximum pool depth. Like the Valleys, the East and West Plains groups differed from one another as the West Plains were saltier, more alkaline, and shallower than the East Plains. The Hills and the Highlands were in the lower-left quadrant and represented a Rocky stream archetype, correlating with rocky substrate and cover and being most strongly associated with riffle channel units. The Hills group tended to have larger substrate and lower nutrient, salinity, and alkalinity values than the Highlands. The Wichita Mountains sites plotted within the Highlands stream group, owing to moderate water chemistry values, large substrate and heterogenous stream morphology. Streams in the Wichita Mountains were considered a part of the Highlands group hereafter.

Compared to the BCA of the six stream groups, the BCA associated with the 13 modified ecoregions showed a much greater degree of overlap. The Central Great Plains and Southwest Tablelands plotted like the West Valleys and West Plains groups described above (Fig. 5). Similarly, the Ouachita Mountains plotted in the lower-left quadrat as the Hills group did, and the Ozark Highlands plotted low on the y-axis like the Highlands stream group that was named for its commonality with the ecoregion. The Arkansas Valley and South-Central Plains occupied the same multivariate space as the East Valleys group. The seven other modified ecoregions spanned across the graph indicating that the modified ecoregions were redundant and contained multiple of the six stream groups identified above.

Figure 5 Between class analysis comparing ecoregions to physicochemical features.

Between class analysis with modified ecoregions compared to habitat and water chemistry. A principal components analysis of habitat and water chemistry features was used to create the multivariate space (A) and the ellipses associated with the modified ecoregions (B).

Anthropogenic effects on stream groups

The BCA comparing anthropogenic stressors revealed differences in land use between the East and West regions, but for the most part, stream groups spanned the range of catchment quality. The upper-left quadrant of the plot was strongly related to hay/pasture and manure, whereas the upper and lower right quadrants were most strongly related to row crop agriculture (Fig. 6). This was an expected difference across the state, as cultivation was more common in the West region compared to the rocky and topographically varied East region. The two western stream groups had near complete overlap in the two right quadrants of the graph indicating that neither are biased by land use. The East Plains group occupied the center of the figure and had nearly equal amounts of hay/pasture and row crop agriculture. The East Valleys and Highlands groups trended more toward hay/pasture land use. The Index of Catchment Integrity and chemical regulation function (CCHEM), that both measure catchment integrity features (Johnson, Liebowitz & Hill, 2019), were positive indicators of catchment condition and primarily explained the y-axis. Only the Hills group tended to have better quality sites and did not span the lower range of catchment integrity. The Hills group was not strongly associated with any land-use activity, and we suspect that this is due to the relatively remote nature of the Hills sites. Although different land uses were apparent among the other stream groups, the distribution of each stream group across the ICI gradient indicated that there were plenty of examples of both good- and poor-quality sites within each group.

Figure 6 Between class analysis comparing stream groups to catchment-scale features associated with anthropogenic activity.

Between class analysis between anthropogenic stressors and stream groups. A principal components analysis of land use and catchment integrity features was used to create the multivariate space (A) and the ellipses associated with the stream groups (B).

Discussion

We accomplished the goal of this study by simplifying reference groups from 13 modified ecoregions to six stream groups while improving explanatory power of the spatial variation in water chemistry and instream habitat. Our six stream groups both represent two regions and exemplify three stream archetypes nested within those regions. The state of Missouri has found a similar hierarchical approach useful for bioassessment. Within Ecological Drainage Units streams are further divided as either glide-pool or riffle-pool segment types (Sarver et al., 2002). Similar archetypes have been described for the STAR project in Europe. Our Rocky (Hills and Highlands) and Valley archetypes resemble the Mountains and Valleys, respectively, although the STAR project described Mediterranean rather than our Plains type (Sandin & Verdonschot, 2006). The archetypes identified in this study framed our empirical data in the context of the hierarchical drivers of species distribution in freshwater stream environments (Cullum et al., 2017). Climate and geology were the ultimate drivers of species distributions, but in climatic and geologic regions that were suitable for a given species, the occurrence of that species depended on the available mesoscale habitat (Frissell et al., 1986; Stevenson, 1997). Further, when both the broad and meso-scale conditions for species occurrence have been met, proximate stressors such as the chemical, physical, or biological contamination may prevent the species from inhabiting the site (e.g., DeLong & Brusven, 1998; Dauwalter et al., 2011). By using the stream archetypes identified in this study, we can account for the natural, broad-scale and meso-scale conditions, and focus on the proximate stressors to stream community. This will allow for more accurate identification and management of poor water-quality causes in watersheds.

The influence of the landscape on stream ecosystems is a core concept in stream ecology; however, previous attempts at attributing conditions in the stream to specific landscape features have been complicated. Alterations in the landscape affect the stream through a chain of context-dependent events that may be mitigated or disrupted resulting in a high degree of complexity between the cause and effect (Allan, 2004). Further, the complexity of landscape-scale features is typically met with multicollinearity making it difficult to distinguish among multiple, potential causes (Legendre, 1993; Graham, 2003; Wagner & Fortin, 2005). In our BCA of anthropogenic stressors, the spatial correlation of the tradeoff between cultivation and grazing with the West to East gradient was apparent. This gradient in land use mirrored the gradient in the natural landscape and illustrated the need to control for spatial variability when assessing the effects of land use on streams. Controlling for natural variability has been accomplished by including continuous, natural features of the landscape in linear models (e.g., King et al., 2005; Anlauf et al., 2011). Defining hypothesis-based models that avoid inclusion of correlated features in the same model (Johnson & Omland, 2004) or reduction of many features into a few principal components using a multivariate analysis are two examples of addressing multicollinearity (Graham, 2003; Purtauf et al., 2005; Wang et al., 2007). The stream groups presented in this study were effectively a categorical representation of the empirically driven principle-components method, whereas the modified ecoregions were more representative of the potentially subjective hypothesis-based approach. The stream groups outperformed the modified ecoregions because the stream groups considered both landscape and instream scales (Cullum et al., 2017). In contrast, modified ecoregions were based only on landscape features in the context of terrestrial ecology. Although the more subjective approach was useful in the past, we believe that the incorporation of the available data in the form of stream groups will better control for the range of natural variability observed in Oklahoma streams.

Description of stream groups

Differences associated with the East and West regions were apparent as a continuum across the stream groups. Although mean annual rainfall was identified as the distinguishing feature between East and West regions, a gradient in water chemistry and land use type was apparent in each analysis. The reduced water availability and habitat complexity in the West resulted in lower species richness compared to the East region (Fausch, Karr & Yant, 1984). Conductivity poses a unique challenge to IBIs in the West region. The effect of conductivity on fishes can be acute as seen when saltwater spills result in fish kills (Liu & Vipulanandan, 2012; Ruiz, 2015), or chronic as when land use activity slowly raises salinity resulting in the loss of intolerant species (Kimmel & Argent, 2009; Zhang, Zhao & Ding, 2019). However, natural saltwater springs exist near the western border of the state raising the possibility of high conductivity obligate species. In the West, the simple view of conductivity as a stressor may be limited when considering that whole ecosystems may be saltwater based. One final stressor expected to affect the West more than the East was the influence of groundwater usage on stream flows. The effects of reduced base flow were detectable in traits-based fish assessments (Santee et al., 2024).

Plains and Valleys archetypes were represented within each of the regions and will provide further nuance to the taxa diversity and richness expectations set by the regions. We expect fish assemblages in the Plains to comprise rheophilic and pelagic-oriented fishes associated with run dominated streams (Johnson, 1942; Matthews, 1988; Worthington et al., 2018), whereas the Valleys will favor detritovores and cover-oriented fishes with a preference for stable, warmer water temperatures (Schlosser, 1982; Bart, 1989; Zorn, Seelbach & Rutherford, 2012). Part of the impetus for this study was the recognition that the difference in centrarchid presence in Great Plains streams was more likely due to habitat than water quality. We were concerned that the Plains archetype within each region reflected streams where row crop agriculture and inadequate riparian buffers had resulted in the fine sediment filling the pools and homogenizing the stream bed (Lenat, 1984; Cooper, 1993; Pusey & Arthington, 2003). However, the third objective of this study indicated that the West Plains and East Plains both spanned the range of catchment condition (ICI) relative to their Valley counterparts, and some streams in the West Plains were in fact among the least anthropogenically influenced sites in the state.

The Highlands and Hills stream groups only occurred in the East region and although they represented a Rocky stream archetype, there were differences in the two groups. Streams in the Hills group had large substrate, more pools, and were generally oligotrophic compared to the Highlands streams (Woods et al., 2005; Splinter et al., 2010). The Highlands streams typically had more ground water input than streams in the Hills (Splinter et al., 2010), which commonly become intermittent in the late summer (Jones & Bergey, 2007; Dyer, Mouser & Brewer, 2016). Both Highlands and Hills streams had elevated levels of endemism within the groups (Mayden, 1985); however, the taxonomic groups most useful to IBIs were represented in each (i.e., sunfish, darters, and suckers; Karr et al., 1986). Highlands streams in the Ozarks have a high diversity of darters and suckers, and the only sculpin species in the state (Griffith, 2003). Highlands streams outside of the Ozarks (e.g., Arbuckle Uplift, Wichita Mountains, and Flint Hills) have lower levels of diversity but are more likely to possess darters and round-bodied suckers than the adjacent, non-Highlands streams (mean count of sensitive benthic species in non-Ozark Highlands = 13, East Plains = 9, and West Plains = 3; OCC data). Like the non-Ozark Highlands streams, the Hills streams possess a subset of the state’s round-bodied suckers and darters and reflect high levels of endemism. Finally, native genotypes of Smallmouth Bass Micropterus dolomieu exist in the Ozarks and Ouachita Mountains ecoregions; however, Smallmouth Bass in the Flint Hills, Wichita Mountains, and Arbuckle Uplift were introduced as sportfish (Brewer & Orth, 2014; Taylor et al., 2018). Anecdotally, the ability of Smallmouth Bass to exist in Highlands and Hills streams across the state supports the grouping of the streams, but further exploration may be needed to determine if the presence of Smallmouth Bass has any impact on fish communities where they were introduced.

Management implications

The categorization of the diversity of stream ecosystems into transparent and reproducible groups supports the development of more precise assessment criteria for biotic indices. Our study used natural stream features that resulted in the recognition of stream groups at two hierarchical levels, regional and stream-reach scales. Verdonschot & Nijboer (2004) found that four hierarchical levels were necessary to create reference conditions for macroinvertebrates across Europe and observed differences in fauna within the Highlands group will likely cause similar challenges in our program. However, the modest survey population in the Highlands group (n = 74), make finer scale groupings unlikely to be useful. Herlihy et al. (2008) addressed this issue by developing Observed/Expected (O/E) models to set site specific reference conditions within aggregated ecoregions. The O/E approach will likely be useful to IBI development, and the stream groups will serve as a control for natural variation at a finer scale (i.e., reach level) than the aggregated ecoregions used by Herlihy et al. (2008). Furthermore, the use of an O/E approach may clarify categorization of streams that exist at the overlapping margin of groups that occur due to the inherently fuzzy nature of categorical delineations of continuous multivariate data (Cullum et al., 2017). Because a subsequent O/E approach would use these stream groups as the foundation to predict species assemblages, poorly predicted streams with marginal group assignment will stand out and we will be able to test the effect of including these streams in other groups.

By considering streams in the hierarchical clustering framework in conjunction with O/E metrics, we will be better positioned to determine when deviation from the typical stream condition expected within a group becomes problematic. The differences in land uses, slopes, soils, and flora within the surrounding landscape all provide context for the expected stressors and the ability of the ecosystems to buffer those stressors. For instance, the threshold for fine sediment as a stressor will likely be lower in the rocky, steep-valley Hills group, than in the West Valleys where alluvial soils and low-gradient fields facilitate crop cultivation. Further, the expected fish assemblage in the West Valley may be negatively affected by a reduction in fine sediment, whereas a similar reduction would positively influence fish assemblages in another stream group (Wohl et al., 2015). As another example, a very salty stream in the West Plains would be expected to have a small but specific assemblage of salt-tolerant fishes resulting in a high O/E score, whereas the assemblage at a slightly salty stream in the East Plains may comprise a diminutive group of only the most tolerant species resulting in a low O/E score. Finally, the effect of land use in the watershed is not immediately apparent but can create legacy effects in the stream that pre-date monitoring programs (Allan, 2004). By having habitat-based stream groups, we can better determine when silty East Valley sites are too silty and investigate patterns in long-term land use that may contribute to the legacy effect of excessive silt (e.g., Anlauf et al., 2011). The creation of the stream groups in this article required the availability of instream habitat data and may not be practical for new programs; however, we argue that existing programs will benefit from incorporating a hierarchical clustering concept into their bioassessment. Given the worldwide application of IBIs, we believe the framework for this analytic approach has relevance for programs well beyond the central United States.

Conclusions

In this study, we successfully reduced 13 modified ecoregions to six stream groups without losing explanatory power of water chemistry and habitat features, or biasing stream groups by anthropogenic activity. The reclassification of streams into hierarchical groups will allow the application of robust reference criteria to sites in several modified ecoregions that were previously under-represented. Although biotic metrics still need to be developed, we expect that these six stream groups will better control for variations in fish assemblages across the state. Species only occur where there is suitable habitat, and the archetypal groups are representative of instream habitat. Nesting stream archetypes within relatively homogenous regions of climate and geology allows for further control of ultimate drivers of species distribution. The Highlands and Hills groups are likely to present some challenges in bio-criteria, because streams within the groups are scattered across the state. We expect similar composition of functional traits among Highlands streams, but species richness will be greatest in the Ozark Highlands. In the Hills group, we saw little evidence of anthropogenic stress at the watershed scale, creating potential difficulty in understanding how streams in the Hills respond to stress. Selecting metrics that are both robust to variation within reference groups and responsive to stressors is always of concern in IBI development, and we expect the same concern to apply to the stream archetypes.

Supplemental Information

Supplemental Information 1 R Script used in the analysis.

Supplemental Information 2 Data used in initial clustering.

Supplemental Information 3 Water chemistry data used to compare clusters.

Supplemental Information 4 Dataset with anthropogenic features use to test for bias among groups.

In this study we relied on the input of Oklahoma Conservation Commission staff who were intimately familiar with the streams assessed. We acknowledge N. Carter, L. Moore, J. Ramming, and W. Shockley for their valuable insight into how well our proposed hierarchical groups represented the conditions they had observed in the field.

Additional Information and Declarations

Competing Interests

The authors declare that they have no competing interests.

Author Contributions

Joseph J. Dyer conceived and designed the experiments, performed the experiments, analyzed the data, prepared figures and/or tables, authored or reviewed drafts of the article, and approved the final draft.

Daniel Dvorett conceived and designed the experiments, analyzed the data, authored or reviewed drafts of the article, and approved the final draft.

Joseph Flotemersch analyzed the data, authored or reviewed drafts of the article, and approved the final draft.

Data Availability

The following information was supplied regarding data availability:

R Script of the analysis used in this work and the raw data are available in the Supplemental Files.

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
