# Peer review of "Using natural landscape and instream habitat to identify stream reference groups for bioassessment"

_PeerJ, doi:10.7717/peerj.20234_

## Round 0.1 · original submission · Major Revisions

Please address all the points raised by the three reviewers.

Reviewer 1 ·

Basic reporting

I think the authors performed a thorough and meaningful analysis of data to produce a grouping of streams for the purposes of ecological monitoring. I found the paper to be engaging and well presented. I have no criticisms to offer with regards to the text of the manuscript.

However, I had a difficult time understanding the figures, especially Figure 1, the map of the ecoregions and proposed stream groups. The stream groups are identified with colored circles laid on top of a map of different colored polygons representing ecoregions, and I found this very distracting. I think the map would be more impactful if the proposed stream groups were identified with different symbols instead of colors. With just 3 commonly used symbols (circle, square, and triangle) in open and closed fashion, the authors could identify all 6 of their proposed stream groups. Perhaps the plains and valleys could be represented by circles and squares with east and west differentiated by open and closed versions of those two symbols. This configuration would better allow the reader to see how stream groups do not always correspond to ecoregion.

For Figure 3, 4, and 5, which are the multivariate plots, I do not understand why the plots of the stream groups are presented twice; once by itself and again in tandem with the environmental factors. I think the tandem plots are sufficient.

Experimental design

No comment

Validity of the findings

No comment

·

Basic reporting

I found the ms easy to read and presented in a professional, concise fashion. The rationale and objective of the study were clearly established in the Introduction, and the Methods clearly defined the approach.

The background literature was good, although overwhelmingly US-centric. This is consistent with the study context being Oklahoma but limits the contribution of the approach to the broader, international bioassessment community.

Table 1 is an efficient presentation of the descriptors used in the various analyses

I would suggest presenting Figure 1 with just Level 3 and 4 ecoregions in the Introduction and then presenting the same scale of map as a new Figure 2 in the Results where only the classifications achieved in the study are shown. It was too confusing (especially for this color-blind reviewer!) to get everything blended together in the current ms Figure 1.

Experimental design

I loved the two-tiered approach and the periodic "reality checks" in the analyses.

The Tier 1 K-means was good, although I would have liked to see more geological descriptors and I wouldn't have used WTDep because of the possibility it has been affected by landuse.

The Tier 2 K-means was more problematic for me because it included several site variables that are either temporally variable, potentially affected by human activity, or both, and my preference for this sort of classification exercise is to only use temporally stable descriptors that are unaffected by humans (except of course climate, which is a more complex topic!). Sticking to temporally stable descriptors also means that each site if just a single observation rather than coming up with a method of summarizing multiple visits to a site (although I admit your method of doing that was pretty cool).

[Note line 294 "principle" should be "principal"]

The BCA also has the same issue as the Tier 2 K-means as water chemistry is both temporally unstable and potentially affected by human activity. This comes up in the "Anthropogenic stressors" analysis. Human activity doesn't happen randomly in natural environments, but the only way to get a handle on the relationship is to rigorously avoid using descriptors that might be affected by human activity.

Validity of the findings

I think the approach was well laid out and justified, and I agree with the interpretation of the classifications. This approach will not only be useful in Oklahoma, but in a variety of contexts internationally. It mixes a strong analytical approach with practicality and that's a great combination for effective management.

Additional comments

I would at least add some discussion recognizing my comments above about using temporally stable decriptors that are unaffected by human activity.

I would also add some material which flags how this is an important contribution beyond the Oklahoma and American context.

Reviewer 3 ·

Basic reporting

This all meets the publication requirements.

Experimental design

Meets basic requirements. Please see additional comments where review is summarized.

Validity of the findings

Meets basic requirements. Please see additional comments where review is summarized.

Additional comments

The paper is well-written, and the methodology and conceptual design are well-described. Based on my assessment, there are a few critical points that should be addressed in the analyses before this paper is published. Attached are my comments by line where primary concerns are described. Overall, here’s a few bullet point highlights from my review:
• Check references and statements where references may be needed. A few instances are highlighted.
• The introduction could be condensed and focus on the impetus of the paper. The authors describe a comprehensive review that could almost be a short review paper of the concepts in itself. Typical journal articles have about half that amount of words and should be able to fit what they need into that frame. Some of the concepts in this section are overlapping such as points about ecoregions and approaches to evaluate them that could be merged.
• The two-tier K-means framework is an interesting approach but still assumes spherical, equal-variance clusters; alternative or complementary methods (hierarchical, model-based clustering) could better capture ecological gradients and nested structure.
• The modest gain over ecoregions warrants statistical tests of group distinctness before claiming stronger explanatory power.
• Hydrologic regime metrics, such as baseflow indices, should be incorporated to help explain east–west gradients and group overlaps, and to strengthen management applications.
• Ordinations suggest some continuous environmental gradients and the authors should consider addressing discussion how fuzzy boundaries affect classification reliability and model performance.

Annotated reviews are not available for download in order to protect the identity of reviewers who chose to remain anonymous.

---

## Round 0.2 · accepted · Accept

You have thoroughly addressed all comments raised by the reviewers.

Reviewer 3 ·

Basic reporting

Authors thoroughly addressed all comments.

Experimental design

Authors thoroughly addressed all comments.

Validity of the findings

Authors thoroughly addressed all comments.

Additional comments

Authors thoroughly addressed all comments.